# Hydrogen Sulfide Metabolite, Sodium Thiosulfate: Clinical Applications and Underlying Molecular Mechanisms

**DOI:** 10.3390/ijms22126452

**Published:** 2021-06-16

**Authors:** Max Y. Zhang, George J. Dugbartey, Smriti Juriasingani, Alp Sener

**Affiliations:** 1Matthew Mailing Center for Translational Transplant Studies, London Health Sciences Center, Western University, London, ON N6A 5A5, Canada; yzha493@uwo.ca (M.Y.Z.); profduu@yahoo.com (G.J.D.); sjuriasi@uwo.ca (S.J.); 2London Health Sciences Center, Multi-Organ Transplant Program, Western University, London, ON N6A 5A5, Canada; 3London Health Sciences Center, Department of Surgery, Division of Urology, Western University, London, ON N6A 5A5, Canada; 4Department of Microbiology & Immunology, Schulich School of Medicine & Dentistry, University of Western Ontario, London, ON N6A 3K7, Canada

**Keywords:** sodium thiosulfate (STS), thiosulfate, hydrogen sulfide (H_2_S), ischemia–reperfusion injury (IRI), sulfide oxidation pathway

## Abstract

Thiosulfate in the form of sodium thiosulfate (STS) is a major oxidation product of hydrogen sulfide (H_2_S), an endogenous signaling molecule and the third member of the gasotransmitter family. STS is currently used in the clinical treatment of acute cyanide poisoning, cisplatin toxicities in cancer therapy, and calciphylaxis in dialysis patients. Burgeoning evidence show that STS has antioxidant and anti-inflammatory properties, making it a potential therapeutic candidate molecule that can target multiple molecular pathways in various diseases and drug-induced toxicities. This review discusses the biochemical and molecular pathways in the generation of STS from H_2_S, its clinical usefulness, and potential clinical applications, as well as the molecular mechanisms underlying these clinical applications and a future perspective in kidney transplantation.

## 1. Introduction

Sodium thiosulfate (STS) is an odorless, inorganic, and water-soluble compound with the chemical formula Na_2_S_2_O_3_ and a molecular weight of 158.11g/mol. It is a major oxidation production of hydrogen sulfide (H_2_S) and is typically available as a white crystalline or powdered substance in the form of pentahydrate (Na_2_S_2_O_3_·5H_2_O) [1]. Currently on the World Health Organization’s list of essential medicines, STS has several other uses including as a common food preservative, a water dechlorinator, a photographic fixative, and a bleaching agent for paper pulp [2]. It possesses therapeutic properties such as antioxidant, anti-inflammatory, and antihypertensive properties [3,4,5,6,7]. It is approved by Food and Drugs Administration (FDA) and is currently clinically useful in the treatment of acute cyanide poisoning, carbon monoxide toxicity, cisplatin toxicities in cancer therapy, and calcific uremic arteriolopathy (calciphylaxis) in dialysis patients [8,9,10,11]. STS is administered intravenously or topically because it is rapidly degraded in the stomach. Emerging reports also suggest its potential application in ischemia–reperfusion injury (IRI) in solid organ transplantation [12,13,14]. In this review, we first present hydrogen sulfide (H_2_S) as an endogenous signaling molecule and a third member of the gasotransmitter family. Next, we describe the biochemical and molecular pathways of H_2_S from which thiosulfate is generated. Finally, we discuss the clinical usefulness and potential clinical applications of STS and its underlying molecular mechanisms, with a future perspective on kidney transplantation.

## 2. Hydrogen Sulfide as a Gasotransmitter

Hydrogen sulfide (H_2_S) is a colorless, flammable, and water-soluble gas with the characteristic smell of rotten eggs [15,16]. For several centuries, H_2_S was notoriously known for its toxic effects and death among agricultural and industrial workers at high concentrations. The mechanism underlying the toxic effect of H_2_S involves reversible antagonism of cytochrome c oxidase (complex IV), the terminal complex of the mitochondrial electron transport chain [17]. In the past two decades, however, this obnoxious-smelling, membrane-permeable gas has risen above its negative public image and is now known to play several important functions in physiological processes at low concentrations. Additionally, it exhibits diverse therapeutic potential with the ability to target several molecular pathways in several diseases and drug-induced toxicities [18,19,20,21,22]. H_2_S is also established among researchers as the third member of the family of gasotransmitters, endogenous gaseous signaling molecules, next to nitric oxide and carbon monoxide [15]. It has the ability to alter activity of proteins from many cellular signaling pathways involved in apoptosis, angiogenesis, inflammation, metabolism, proliferation, and oxygen sensing. It can also play a detoxifying role during oxidative stress by increasing the development of glutathione [23,24,25], the most abundant naturally occurring antioxidant in the body, and by reacting directly with peroxynitrite (ONOO^−^) as a direct scavenging property of H_2_S toward cellular ROS. H_2_S is endogenously produced in all mammalian cells through metabolic pathways that use the sulfur-containing amino acid L-cysteine and 3-mercaptopruvate via 3 enzymes: cystathionine β-synthase (CBS), cystathionine γ-lyase (CSE), and 3-mercaptopyruvate sulfurtransferase (3-MST) (Figure 1). It has also been found that H_2_S can be produced from D-cysteine using the peroxisomal enzyme, D-amino acid oxidase [26]. Besides its endogenous production, H_2_S is also administered exogenously through a number of its donor compounds, including STS and GYY4137 [27,28,29].

### 2.1. Generation of STS from H_2_S

At a physiological level, thiosulfate can be generated in tissues from the mitochondrial sulfide oxidation pathway, using H_2_S as the substrate. This process involves three mitochondrial enzymes: quinone oxidoreductase, sulfur dioxygenase, and sulfur transferase (Figure 1). Using an isolated mitochondria rat model, Hildebrandt et al. [30] proposed a method into the biochemical pathway of H_2_S oxidation to thiosulfate. Firstly, as illustrated in Figure 1, H_2_S can react with a membrane-bound disulfide on quninone oxidoreductase (SQR) to generate a membrane-bound persulfide group (SQR-SSH). A persulfide dioxygenase in the mitochondrial matrix oxidizes one persulfide molecule to sulfite (H_2_SO_3_), which is then used in a sulfurtransferase reaction catalyzed by the enzyme rhodanase to form thiosulfate [30]. Rhodanase is a mitochondrial enzyme that transfers a sulfur atom from sulfane-containing donor to the thiophilic acceptor substrate [31]. The catalytic activity of rhodanase occurs via a double displacement mechanism, where the active site, a cysteine residue (Cys247), accepts a sulfur atom from the persulfide intermediate state, followed by the transfer of sulfide sulfur from the enzyme to the nucleophilic acceptor sulfite, which produces thiosulfate (Figure 1) [32]. Although human mitochondria also utilize this sulfide oxidation pathway, recent evidence suggests that glutathione (GSH) functions as a persulfide acceptor for human SQR to produce the persulfide intermediate [33,34]. Most recently, Libiad et al. [35] found that the kinetic behavior of these enzymes favors SQR by using GSH as an acceptor to form glutathione persulfide (GSSH), which is then converted to thiosulfate by human rhodanase (Figure 1). This is further confirmed by kinetic simulations in previous rat liver mitochondria studies with or without GSH, which supports GSSH as the first intermediate formed in the flow of the sulfide oxidation pathway [34].

After learning that H_2_S can generate thiosulfate via the sulfide oxidation pathway, it is important to understand that the reverse reaction also occurs at a physiological level in tissue. In a study using recombinant human SQR in *Escherichia coli*, Jackson et al. [33] showed that the metabolism of thiosulfate is catalyzed by thiosulfate reductase, as it consumes two GSH molecules and results in the generation of sulfite, oxidized glutathione, and H_2_S. Further evidence of the ability of thiosulfate to produce H_2_S via a glutathione-dependent reduction was confirmed by a study in which exogenous thiosulfate treatment significantly decreased GSH/GSSG ratio to total sulfide ratio in a dose-dependent manner [36]. In addition, Olson et al. [37] found that H_2_S generation from thiosulfate can also occur under the presence of 1,2-Dithiole-3-thiones, an exogenous reducing agent. However, regardless of the exact mechanism of how sulfur is transferred, thiosulfate appears to be a key intermediate. Thus, thiosulfate (in the form of STS) is a major oxidation product of H_2_S.

### 2.2. Biological Properties of Thiosulfate

In addition to thiosulfate being a stable, nontoxic metabolite of H_2_S [38], it is also a sulfane sulfur, which is defined as sulfur atoms covalently bonded to other sulfur atoms, making it unstable and readily oxidizing in air and reducing with thiols [39,40]. Compounds containing sulfane sulfur are known to possess cell regulatory effects through the activation or inactivation of enzymes and changing protein activities [41,42]. The functions of sulfane sulfur include antioxidant regulation, tRNA sulfuration, and iron-sulfur protein formation [41,43,44]. The ability of mitochondrial enzymes to generate thiosulfate from H_2_S and vice versa could have misinterpretations on which sulfur molecule conducts the biological signalling. In a mouse model of heart failure, Sen et al. [12] demonstrated that 3 mg/mL of oral thiosulfate can increase depleted H_2_S levels. In addition, Tokuda et al. [45] observed the impact of H_2_S gas on lipopolysaccharide (LPS)-induced inflammation in mice. They found that H_2_S inhalation after LPS challenge increased plasma thiosulfate level and rhodanase activity, which prevented LPS-induced inflammation. The authors’ opinion that thiosulfate may contribute to beneficial effects of H_2_S inhalation was verified after they found that administering thiosulfate improved survival after LPS challenge in a dose-dependent manner. This suggests that it is thiosulfate, not H_2_S, that participates as a signalling molecule in cellular regulatory processes [46].

## 3. Clinical Usefulness of STS

### 3.1. STS in the Treatment of Cyanide Poisoning

The clinical usefulness of STS was first identified in the 1930s, when it was co-administered with sodium nitrite to treat acute cyanide poisoning. As cyanide is a cytotoxic agent that binds to cytochrome oxidase and thereby inhibits cellular respiration, STS serves as a donor of sulfur, which is used as a substrate by the enzyme rhodanase (a sulfur transferase) to covert cyanide to thiocyanate, a nontoxic cyanide molecule, which is then excreted in urine [8,47]. This occurs after sodium nitrate removes cyanide from the mitochondrial electron transport chain by inducing the formation of methemoglobin [8,47]. Since then, there has been several pieces of evidence in preclinical and clinical studies validating STS as an antidote to cyanide poisoning and other chemical poisoning, such as carbon monoxide–cyanide toxicity in patients for whom sodium nitrite is contraindicated [48,49]. The United States, for example, has a standard cyanide antidote kit, which first uses 10mL intravenous sodium nitrite, followed immediately by 50mL intravenous STS [50]. Taken together, these results show that the mechanism of action underlying the use of STS as an antidote to cyanide poisoning is due to its ability to serve as a sulfur donor.

### 3.2. STS in the Treatment of Cisplatin Toxicities in Cancer Therapy

Besides being an antidote to cyanide poisoning, STS is also a neutralizing agent that protects against cisplatin toxicity. Cisplatin is one of the most widely used agents to treat solid tumors. However, it has adverse effects on renal, auditory, neurological, and hematological systems [9]. Laplace et al. [51] showed that administering STS protects against renal impairment following cisplatin chemotherapy. Moreover, administration of high-dose cisplatin over the last 2 h of STS infusion prevented possible cisplatin-induced nephrotoxicity, as there were no observed changes in elimination rate constant, volume distribution and total body clearance compared to patients who received low-dose cisplatin without STS [52]. This observation suggests that STS allows higher doses of cisplatin to be administered before dose-limiting toxicity occurs. This protection is thought to be related to STS binding to free platinum, resulting in total clearance of inactive metabolite and limiting renal tubular cell necrosis [53]. In addition, the use of cisplatin to effectively treat childhood hepatoblastoma can cause severe and permanent ototoxicity, leading to eventual hearing loss. Interestingly, treatment with STS 6 h after cisplatin chemotherapy resulted in a lower incidence of cisplatin-induced hearing loss among children with standard-risk hepatoblastoma without jeopardizing overall and event-free survival [54]. Furthermore, STS was shown to inhibit oxidative stress-induced ototoxicity in the cochlea [54]. Following these positive outcomes and after two successful clinical trials, Fennec Pharmaceuticals is currently waiting on FDA approval on the first potential prevention of platinum induced-ototoxicity in pediatric patients.

### 3.3. STS in the Treatment of Calciphylaxis in Dialysis Patients

The clinical usefulness of STS has grown over the years to include treatment of calciphylaxis, which is a severe complication in patients with advanced chronic kidney disease in which calcium accumulates in blood vessels [27,55,56]. Predominantly seen in people with end-stage kidney disease, calciphylaxis is a predictor of cardiovascular death in long-term hemodialysis patients [57]. It is characterized by systemic medial calcification of the arterioles, leading to ischemia and subcutaneous necrosis. Promising results have been obtained through the use of intralesional STS. Areas of clinically active disease were treated with 250 mg/mL STS, resulting in the resolution of calciphylaxis lesions over a period of weeks with no recurrence of the disease [9]. Most recently, Peng et al. [11] conducted a systematic review of several cases on the use of STS for calciphylaxis and found that STS has a promising role as an effective therapy for calciphylaxis by acting as a calcium-chelating agent, binding to Ca^2+^ and increasing its solubility. The authors also reported that STS possesses vasodilatory and antioxidant properties. Their findings were in agreement with previous reports that suggested that STS could combine with insoluble tissue calcium salts to form calcium thiosulfate, a salt that can later be dialyzed [11,58,59,60]. Thus, treatment of calciphylaxis with STS is partly due to its antioxidant and calcium-chelating and vasodilatory properties.

## 4. Potential Clinical Applications of STS

### 4.1. STS in the Treatment of Renovascular Hypertension

As an H_2_S donor molecule, STS is thought to have unexplored therapeutic potential in the context of many diseases. Over the past few years, a number of independent groups have discovered the beneficial effects of STS in animal models of disease (Table 1). For example, a recent study examined the protective properties of STS in angiotensin II-induced renovascular hypertension in rats [4]. The authors observed that 1 g/kg dose of STS treatment per day induced a lower plasma urea, proteinuria, and improved creatinine clearance through its antioxidant property. They also attributed the protective effect of STS partly to its anti-inflammatory property, preventing angiotensin II-induced influx of macrophages [4]. This finding supports several previous reports that highlighted anti-inflammatory property of STS in downregulating pro-inflammatory genes such as IL-1β, TNF-α, and MAP-1 and reduced macrophage recruitment [5,61,62]. In a recent experimental rat model of hyperoxaluria and renal injury, 0.4 g/kg dose of STS treatment scavenged reactive oxygen species (ROS) in a dose-dependent manner, mitigated cellular hydrogen peroxide levels, and maintained superoxide dismutase activity [6]. It is important to note that thiosulfate has two lone electron pairs: one at the single bonded sulfur moiety of the disulfide bond and the other at the single bonded oxygen [60]. This characteristic allows thiosulfate to act as an effective antioxidant by donating electrons to unpaired damaging electrons associated with mitochondrial ROS [37,63,64]. Further evidence of the antioxidant property of thiosulfate was confirmed in a mouse model of congestive heart failure by Sen et al. [12], where they reported that thiosulfate scavenged superoxide in myocardial tissue. In addition, thiosulfate can react with superoxide to form glutathione, a thiol-dependent antioxidant system in mammalian cells [3]. In conclusion, STS possesses potent antioxidant and anti-inflammatory properties, which protect against renovascular hypertension and other models of renal injury.

### 4.2. STS in Ischemia–Reperfusion Injury

An additional area in which STS has been reported to show protective effects is in animal models of ischemia–reperfusion injury (IRI) (Table 1). IRI is defined as tissue injury due to temporary cessation of blood flow (ischemia) and subsequent restoration of blood flow to the ischemic tissue [72]. Chronic inflammation, excessive ROS production, ATP depletion, accumulation of succinate, and induction of cellular apoptotic pathways are major molecular events associated with IRI [73,74,75]. The first major molecular event of IRI occurs when cells are deprived of adequate oxygen due to cessation of blood flow. Lack of oxygen results in energy depletion, since the cells are unable to synthesize ATP [76]. The depletion of ATP causes a rise in inorganic phosphate and inhibition of Na^+^/K^+^ pumps, resulting in increased intracellular Ca^2+^ concentration and mitochondrial inner membrane permeability [77]. Additionally, prolonged ischemic time can damage multiple complexes in the electron transport chain (ETC), causing it to be more prone to electron leakage [78]. The second molecular event in IRI occurs when blood flow is restored to the ischemic tissue. Reperfusion is often characterized by increased formation of ROS, decreased ATP production, and cell death [79]. Previous studies have shown that overproduction of ROS occurs from the mitochondrial ETC when oxygen is reintroduced to the cell, with the oxygenation of succinate as a main superoxide generating species via reverse electron transport [74,75,79]. ROS can damage proteins of the ETC complexes, which further inhibits ATP production and increases electron leakage [80]. Cellular ATP depletion initiates translocation of pro-apoptotic proteins such as BAX, which causes mitochondrial swelling and induces efflux of cytochrome c and apoptosis-inducing factor [81]. These factors in turn activate caspase 3 apoptotic signalling cascade, initiating cellular apoptosis.

In an experimental model of renal IRI, in which isolated rat mitochondria were subjected to physiological oxidative stress by nitrogen gas purging, treatment with STS induced renal protection and maintained mitochondrial functional integrity by markedly reducing oxidative stress and deteriorated mitochondrial enzyme activities compared to untreated groups [14]. In addition, Ravindran et al. [70] observed in in vitro and in vivo models of cardiac IRI that STS preconditioning significantly increased NADH dehydrogenase activity and decreased oxidative stress due to increased activities of antioxidant enzymes such as catalase and superoxide dismutase. The authors further showed in an in silico model that STS has higher binding affinity for caspase-3 because of its perfect fit in the active site Cys-163, which is stabilized via strong hydrogen bonds. This results in inactivation of caspase-3 by preventing access of natural substrate to caspase-3 binding site, ultimately halting apoptosis [70]. Further evidence of the higher binding affinity of STS for caspase-3 was confirmed in a mouse cerebral IRI study by Marutani et al. [65], where the authors observed that 10 mg/kg dose of STS inhibits caspase-3 activity via persulfidation of the same active site, Cys-163, and protects against neuronal IRI in mice. Additionally, STS was shown to activate Erk 1/2 and block the c-Jun N-terminal kinase (JNK), which led to inhibition of apoptosis by preventing dephosphorylation of the pro-apoptotic protein, Bad, and downregulation of anti-apoptotic protein, Bcl-2 [82,83,84].

The protective effect STS observed in different experimental models of IRI is partly attributable to its ability to activate mitochondrial adenosine triphosphate (ATP)-sensitive potassium (K_ATP_) channels, suggesting that the opening of these channels may have inhibited mitochondrial permeability transition [12,13]. This observation confirms previous studies in which H_2_S stimulated the opening of K_ATP_ channels by blocking phosphorylating of the transcription factors fork head box O (FOXO1 and FOXO3a) in rat vascular smooth muscle cells, leading to reduced Ca^2+^ influx and preventing the opening of mitochondrial permeability transition pores [85,86]. Further evidence of STS maintaining mitochondrial integrity was confirmed in a study by Mohan et al. [14], where isolated rat mitochondria were subjected to physiological oxidative stress. The results showed that the pretreated STS group had higher renal mitochondrial enzyme activity due to its increased NADH dehydrogenase activity compared to the nontreated group [14]. More recently, Ravindran et al. [69] reported in a rat model of cardiac IRI that preconditioned with 1 mM STS exhibited similar ATP synthase activity and mitochondrial enzyme activity compared to sham [69]. The same authors conducted another rat model of cardiac IRI study, but this time, the rats were postconditioned with 1 mM STS [68]. The STS-treated group improved activities of mitochondrial ETC complex enzymes I-IV and showed significantly increased expression of PGC-1α, a positive regulator of mitochondrial biogenesis, ATP production, and ROS-detoxifying system [68]. In summary, STS modulates several molecular pathways in the mitochondria, leading to protection against IRI in various tissues.

## 5. Future Direction

### 5.1. Future Direction in the Use of STS as an H_2_S Donor Molecule

Considering that STS is already a clinically viable H_2_S donor drug approved by FDA and is also in clinical trials along with other H_2_S donor drugs such as GIG-1001, SG1002, ATB-436, and Zofenopril for cardiovascular diseases, intestinal disorders, and other conditions, [87] it is important to translate these promising experimental findings about STS to clinical practice. As such, STS-related therapeutic research is a rapidly emerging field, with many studies done on H_2_S-related cytoprotective effects. One example is signaling mechanism of the antioxidant and transcription factor nuclear factor erythroid-related factor 2 (Nrf2), which is partly attributable to H_2_S effects [88] (Figure 2). Previous studies have shown that H_2_S activates Nrf2-dependent signaling, which produces antioxidant proteins to mitigate animal models of inflammatory acute liver failure and cardiovascular disease [89,90,91,92]. Under normal conditions, Nrf2 is captured by Keap1 proteins in the cytoplasm [93]. However, when exposed to oxidative stress, Nrf2 avoids Keap1 and is translocated into the nucleus in order to bind to antioxidant response elements (ARE) to induce expression of various antioxidant gene clusters [88,92,94]. In a recent study by Koike et al. [95,96], they discovered that addition of sulfane sulfurs increased Nrf2 accumulation in the nucleus of neuroblastoma cells through the structural change of Keap1 protein. Specifically, the sulfane sulfurs triggered a persulfidation reaction of the cysteine residue in Keap1, which led to Keap1 forming homodimers with another Keap1 protein or heterodimers with another protein (Figure 2). It has also been reported that persulfidated proteins are protective against oxidative stress-induced damage and thereby preserve the function of the persulfidated cysteine residues [97]. The authors further reported that the polysulfide induced AKT phosphorylation, which triggered the phosphorylation of Nrf2, resulting in nuclear translocation. This suggests that sulfane sulfurs, such as thiosulfate, activate Nrf2 signaling pathway through the structural change of Keap1 protein and phosphorylation of AKT (Figure 2). These findings support a recent study about a mice model of acute liver failure, where administration of 2 g/kg STS attenuated liver injury by enhancing AKT phosphorylation and inducing Nrf2-dependent antioxidant proteins [21]. In addition, the authors showed that STS treatment also inhibited phosphorylation of JNK, a protein that is upregulated by inflammatory cytokines and extracellular stresses and plays a critical role in apoptotic signaling [98]. A proposed mechanism of how thiosulfate interacts with Nrf2 signaling pathway to induce cytoprotective effects against oxidative stress is shown in Figure 2.

Many of the biochemical characteristics of H_2_S signaling that provide cytoprotective effects, such as persulfidation of signaling proteins, can be accomplished with thiosulfate instead. For example, a study by Giovinazzo et al. [99] showed that H_2_S donor molecule GYY4137 inhibits Tau hyperphosphorylation by persulfidation of kinase GSK3β, ultimately ameliorating cognitive and motor deficits in Alzheimer’s disease. STS, as we previously mentioned, has been shown to trigger persulfidation reactions in the sulfur oxidation pathway [37] and in the Nrf2 signaling pathway [95]. Multiple studies on H_2_S prodrugs reported to be beneficial in cardiovascular systems also involve similar biochemical mechanism to STS, such as persulfidation in SP1-medated transcription to preserve endothelial function [100,101], persulfidation in regulation of PYK2-mediated eNOS phosphorylation to mediate cardioprotection [102], and persulfidation in ERK/MEK1/PARP-mediated DNA damage repair and cell survival [103]. It is possible but remains to be studied whether modulation of STS pathways may contribute to the therapeutic actions of the experimental H_2_S prodrugs. Since STS is widely available as a key metabolite of H_2_S with similar biochemical signaling effects and as a clinically approved drug, further studies are warranted on the protective effects of STS in central nervous system, cardiovascular system, and many other system pathologies.

### 5.2. Future Direction in the Use of STS in Organ Transplantation

Following our compelling success in different animal models of kidney transplantation with the use of H_2_S-supplemented University of Wisconsin (UW) solution for static cold storage, it is important to translate these promising experimental findings to clinical practice using STS. Hence, we decided to investigate whether STS-supplemented UW solution would be suitable for renal graft preservation. In recent rat syngeneic transplant experiments, we found that kidneys stored and perfused with STS-supplemented UW solution showed better survival, improved acute tubular necrosis scores, and improved graft function compared to UW-stored kidneys without STS treatment [104]. Serum creatinine levels also showed that, while STS-treated rats exhibited significantly decreased serum creatinine at postoperative day 3 compared to UW-treated rats (without STS), serum creatinine levels in the former group were not statistically different from that of sham-operated rats [104]. Overall, supplementing organ preservation solutions with STS may be a promising approach, as it requires minimal modification of existing clinical protocols and is also cost-effective. However, mechanistic properties of STS on renal IRI need to be studied further.

## 6. Conclusions

Although H_2_S is a major contributor to altering cellular physiology in various ways, STS appears to play a significant role in biological signaling as well. Several studies have elucidated the ability of mitochondrial enzymes to generate thiosulfate from H_2_S through a sulfide oxidation pathway. Emerging data on the biological effects of STS and its close chemical relationship with H_2_S support the development of STS-based therapeutics. Besides its clinical usefulness, STS has also been shown experimentally to effectively protect against renovascular hypertension and other models of renal injury. In the context of kidney transplantation, modification of the preservation solutions with STS may be a simple, inexpensive, and nontoxic novel therapeutic strategy to mitigate cold IRI in donor organs to ultimately improve graft outcomes and minimize posttransplant complications. However, the underlying protective molecular mechanisms of this novel approach will need further investigation.

## Figures and Tables

**Figure 1 ijms-22-06452-f001:**
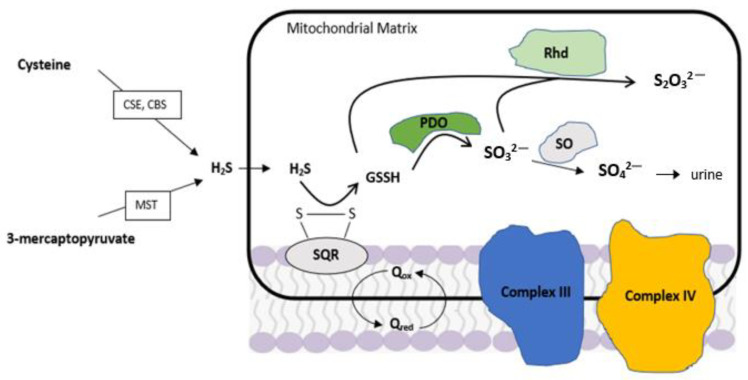
Generation of thiosulfate from H_2_S in the mitochondrial sulfide oxidation pathways. Hydrogen sulfide (H_2_S) is produced by enzymes cystathione γ-lyase (CSE) and cystathionine β-synthase (CBS) in the trans-sulfuration pathway. A third enzyme, 3-mercaptopyruvate sulfurtransferase (MST), also produces endogenous H_2_S in the presence of the substrate 3-mercaptopyruvate. A membrane-bound sulfide, quinone oxidoreductase (SQR), oxidizes H_2_S to persulfide, which is transferred to a glutathione (GSH). A persulfide dioxygenase (PDO) in the mitochondrial matrix oxides one glutathione persulfide (GSSH) to sulfite (H_2_SO_3_), which is then used in a sulfurtransferase reaction catalyzed by the enzyme rhodanase (Rhd) to form thiosulfate (S_2_O_3_^2−^) by transferring a second glutathione persulfide from SQR to sulfite. Sulfite can be further oxidized by sulfite oxidase (SO) to form sulfate (SO_4_^2−^) and is subsequently excreted in urine. PDO and SO are oxygen dependent enzymes.

**Figure 2 ijms-22-06452-f002:**
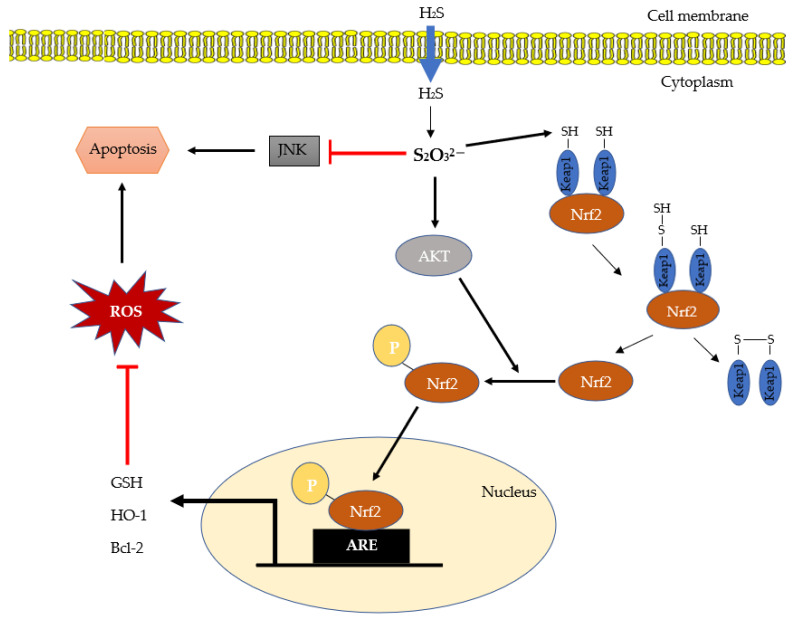
Proposed overview of cytoprotective effects of thiosulfate against oxidative stress. Thiosulfate (S_2_O_3_^2−^) is produced from hydrogen sulfide (H_2_S) via sulfide oxidation pathway. The bound sulfur on thiosulfate activates the Nrf2 system through the structural change of Keap1 proteins and induction of phosphorylated AKT. The nuclear translocation of phosphorylated Nrf2 binds to ARE to promote expression of various antioxidative gene clusters. Thiosulfate also contributes to anti-apoptotic signaling via inhibition of JNK phosphorylation. ROS, reactive oxygen species; GSH, glutathione; HO-1, heme oxygenase-1; Bcl-2, B-cell lymphoma-2; ARE, antioxidant response element; Nrf2, nuclear factor erythroid-related factor 2; Keap1, Kelch-like ECH-associated protein 1; AKT, protein kinase B; JNK, c-Jun N-terminal kinases.

**Table 1 ijms-22-06452-t001:** Summary of mechanisms of action of STS in animal models of human diseases.

Experimental Model	STS Concentration	Effect of STS	References
BCAO-induced cerebral IRI in mice	10 mg/kg	- Improved neurological function and survival - Inhibitedcaspase-3 activity - Mitigated apoptosis via JNK blocking	[65]
AVF-induced heart failure in mice	3 mg/mL	- Protected against cardiac dysfunction - Elevated endogenous production of H_2_S - Prevented the increase in MMP-2, MMP-9, and TIMP-1 expression levels	[12]
Hyperoxaluria in rats	0.4 g/kg/b.w.t	- Preserved superoxide dismutase activity	[6]
Ethylene glycol-induced nephrolithiasis in rats	400 mg/Kg b.w.t	- Increased renal protection by modulating the mitochondrial KATP channel - Showed normal serum creatinine and renal tissue architecture	[66]
Angiotensin II-induced heart disease in rats	1 g/kg/day	- Attenuated hypertensive cardiac disease - Regulated blood pressure - Reduced ANP mRNA levels	[45]
Angiotensin II-induced hypertension, proteinuria, and renal damage in rats	1 g/kg/day	- Increased GSH levels - Reduced influx of macrophages to near-control levels - Improved creatinine clearance	[4]
L-NNA-induced hypertension in rats	2 g/kg/day	- Enhanced GFT and ERPF - Protected against glomerulosclerosis - Lowered plasma urea and renal vascular resistance	[67]
Myocardial IRI in rats	1 mM (Postconditioned)	- Reduced myocardial infarct size - Lowered expression of caspase-3 and PARP	[68]
Renal mitochondrial IRI in rats	400 mg/kg	- Maintained mitochondrial function - Increased NADH hydrogenase activity	[14]
Myocardial IRI in rats	1 mM (Preconditioned)	- Preserved mitochondrial ATP synthesis - Increased PGC-1α expression - Improved ETC complex enzyme activities	[69]
LAD occlusion model of cardiac IRI in rats	0.1–1 mM	- Reduced apoptosis associated with mitochondrial dysfunction - Lowered levels of cardiac injury markers LHD and CK	[70]
Cardiac IRI with PAG in rats	1 mM	- Preserved protective mechanisms in presence of PAG	[71]
GalN/LPS-induced liver injury in mice	2 g/kg	- Increased Nrf2 and Akt-dependent signaling - Inhibited JNK phosphorylation	[21]

BCAO, bilateral common carotid artery occlusion; STS, sodium thiosulfate; JNK, c-Jun N-terminal kinase; IRI; ischemia–reperfusion injury; GSH, glutathione; AVF, arteriovenous fistula; MMP, matrix metalloproteinases; TIMP, tissue inhibitors of matrix metalloproteinases; ANP, atrial natriuretic peptide; GFR, glomerular filtration rate; L-NNA, N-u-nitro-L-arginine; ERPF, effective renal plasma flow; ETC, electron transport chain; LAD, left anterior descending artery; LHD, lactate dehydrogenase; CK, creatine kinase; PAG, D, L-propargylglycine; GalN, D-galactosamine; LPS, lipopolysaccharide; Nrf2, nuclear factor erythroid related-factor 2.

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
