# Peer review of "Hydrogen Sulfide Metabolite, Sodium Thiosulfate: Clinical Applications and Underlying Molecular Mechanisms"

_ijms, 2021, doi:10.3390/ijms22126452_

Round 1

Reviewer 1 Report

1. Sentence 33; Add the following reference to the 3-6 reference list: Roorda, M., Miljkovic, J.L., van Goor, H., Henning, R.H., and Bouma, H.R. (2021). Spatiotemporal regulation of hydrogen sulfide signaling in the kidney. Redox Biol. 43, 101961.

2. Sentence 61; Extend the explanation regarding the antioxidative effects of H2S by commenting its direct reaction with peroxynitrite (ONOO.-), RSNO and superoxide as a direct scavenging properties of H2S towards main cellular ROS and RNS species. As well, add the following references to the reference 22: Whiteman, M., Armstrong, J.S., Chu, S.H., Jia-Ling, S., Wong, B.-S., Cheung, N.S., Halliwell, B., and Moore, P.K. (2004). The novel neuromodulator hydrogen sulfide: an endogenous peroxynitrite “scavenger”? J. Neurochem. 90, 765–768. Filipovic, M.R., Miljkovic, J., Allgäuer, A., Chaurio, R., Shubina, T., Herrmann, M., and Ivanovic-Burmazovic, I. (2012). Biochemical insight into physiological effects of Hâ‚‚S: reaction with peroxynitrite and formation of a new nitric oxide donor, sulfinyl nitrite. Biochem. J. 441, 609–621. 3. Sentence 75; Add appropriate indication which of the H2S catabolism enzymes are oxygen dependent (PDO and SO) in the Figure 1. 4. Sentence 122; Add the following reference: Wood, J.L. (1987). Sulfane sulfur. Methods Enzymol. 143, 25–29. 5. Sentence 124; Add the following reference as well: Mustafa, A.K., Gadalla, M.M., Sen, N., Kim, S., Mu, W., Gazi, S.K., Barrow, R.K., Yang, G., Wang, R., and Snyder, S.H. (2009). HS signals through protein S-Sulfhydration. Sci. Signal. 6. Sentence 213; This sentence is not delivering a clear information. Please format this sentence. 7. Sentence 226; Add the Ischemic accumulation of succinate as an important process during the ischemic phase.
Add the following reference:

Chouchani, E.T., Pell, V.R., Gaude, E., Aksentijević, D., Sundier, S.Y., Robb, E.L., Logan, A., Nadtochiy, S.M., Ord, E.N.J., Smith, A.C., et al. (2014). Ischaemic accumulation of succinate controls reperfusion injury through mitochondrial ROS. Nature 515, 431–435. Chouchani, E.T., Pell, V.R., James, A.M., Work, L.M., Saeb-Parsy, K., Frezza, C., Krieg, T., and Murphy, M.P. (2016). A Unifying Mechanism for Mitochondrial Superoxide Production during Ischemia-Reperfusion Injury. Cell Metab. 23, 254–263.   8. Sentence 230; Add the oxygenation of succinate as a main superoxide generating specie via RET.

Add the following references: Chouchani, E.T., Pell, V.R., Gaude, E., Aksentijević, D., Sundier, S.Y., Robb, E.L., Logan, A., Nadtochiy, S.M., Ord, E.N.J., Smith, A.C., et al. (2014). Ischaemic accumulation of succinate controls reperfusion injury through mitochondrial ROS. Nature 515, 431–435. Chouchani, E.T., Pell, V.R., James, A.M., Work, L.M., Saeb-Parsy, K., Frezza, C., Krieg, T., and Murphy, M.P. (2016). A Unifying Mechanism for Mitochondrial Superoxide Production during Ischemia-Reperfusion Injury. Cell Metab. 23, 254–263. 9. Sentence 248;
It is not clear which antioxidant enzymes are persulfidated?
In addition to reference 66 and 67 as well as the corresponding sentences add and comment the metalloprotein catalysed persulfidation of caspase by H2S as an important link between mitochondrial sulfide catabolism / thiosulfate anabolism and persulfidation of caspase enzymes affecting the apoptotic processes.

10. Sentence 307;
Add the following reference along the reference 83:

Zivanovic, J., Kouroussis, E., Kohl, J.B., Adhikari, B., Bursac, B., Schott-Roux, S., Petrovic, D., Miljkovic, J.L., Thomas-Lopez, D., Jung, Y., et al. (2019). Selective Persulfide Detection Reveals Evolutionarily Conserved Antiaging Effects of S-Sulfhydration. Cell Metab. 30, 1152-1170.e13.

11. Sentence 311; Check the reference 84!

Author Response

Please see the attachment: 

  1. Manuscript with track changes according to the reviewer's comments.

First of all, I would like to kindly thank the reviewers for their feedbacks on our manuscript and thereby, assisting in improving the quality of our work. Please find our responses to each comment below in dark blue text.

Reviewer #1

“1. Sentence 33; Add the following reference to the 3-6 reference list: Roorda, M., Miljkovic, J.L., van Goor, H., Henning, R.H., and Bouma, H.R. (2021). Spatiotemporal regulation of hydrogen sulfide signaling in the kidney. Redox Biol. 43, 101961.”

This has been added as reference 7 in the revised manuscript.

“2. Sentence 61; Extend the explanation regarding the antioxidative effects of H
2S by commenting its direct reaction with peroxynitrite (ONOO.-), RSNO and superoxide as a direct scavenging properties of H2S towards main cellular ROS and RNS species. As well, add the following references to the reference 22: Whiteman, M., Armstrong, J.S., Chu, S.H., Jia-Ling, S., Wong, B.-S., Cheung, N.S., Halliwell, B., and Moore, P.K. (2004). The novel neuromodulator hydrogen sulfide: an endogenous peroxynitrite “scavenger”? J. Neurochem. 90, 765–768. Filipovic, M.R., Miljkovic, J., Allgäuer, A., Chaurio, R., Shubina, T., Herrmann, M., and Ivanovic-Burmazovic, I. (2012). Biochemical insight into physiological effects of Hâ‚‚S: reaction with peroxynitrite and formation of a new nitric oxide donor, sulfinyl nitrite. Biochem. J. 441, 609–621.

As suggested by the reviewer, we have extended the explanation of the antioxidant effect of H2S by stating that reacts directly with peroxynitrite (ONOO-) as a direct scavenging property of H2S towards cellular ROS.

Also, we have added the recommended references as references 24 and 25 in the revised manuscript.

“3. Sentence 75; Add appropriate indication which of the H2S catabolism enzymes are oxygen dependent (PDO and SO) in the Figure 1.”

This has been added to figure 1 accordingly

“4. Sentence 122; Add the following reference: Wood, J.L. (1987). Sulfane sulfur. Methods Enzymol. 143, 25–29.”

This has been added as reference 40 in the revised manuscript.

“5. Sentence 124; Add the following reference as well: Mustafa, A.K., Gadalla, M.M., Sen, N., Kim, S., Mu, W., Gazi, S.K., Barrow, R.K., Yang, G., Wang, R., and Snyder, S.H. (2009). HS signals through protein S-Sulfhydration. Sci. Signal.”

This has been added as reference 42 in the revised manuscript.

“6. Sentence 213; This sentence is not delivering a clear information. Please format this sentence.”

We have rephrased the sentence

“7. Sentence 226; Add the Ischemic accumulation of succinate as an important process during the ischemic phase.

Add the following reference:

Chouchani, E.T., Pell, V.R., Gaude, E., Aksentijević, D., Sundier, S.Y., Robb, E.L., Logan, A., Nadtochiy, S.M., Ord, E.N.J., Smith, A.C., et al. (2014). Ischaemic accumulation of succinate controls reperfusion injury through mitochondrial ROS. Nature 
515, 431–435. Chouchani, E.T., Pell, V.R., James, A.M., Work, L.M., Saeb-Parsy, K., Frezza, C., Krieg, T., and Murphy, M.P. (2016). A Unifying Mechanism for Mitochondrial Superoxide Production during Ischemia-Reperfusion Injury. Cell Metab. 23, 254–263.” 

We have added the recommended references as references 69 and 70 in the revised manuscript and also added “accumulation of succinate” as process during ischemic phase.

“8. Sentence 230; Add the oxygenation of succinate as a main superoxide generating specie via RET.

Add the following references: Chouchani, E.T., Pell, V.R., Gaude, E., Aksentijević, D., Sundier, S.Y., Robb, E.L., Logan, A., Nadtochiy, S.M., Ord, E.N.J., Smith, A.C., et al. (2014). Ischaemic accumulation of succinate controls reperfusion injury through mitochondrial ROS. Nature 
515, 431–435. Chouchani, E.T., Pell, V.R., James, A.M., Work, L.M., Saeb-Parsy, K., Frezza, C., Krieg, T., and Murphy, M.P. (2016). A Unifying Mechanism for Mitochondrial Superoxide Production during Ischemia-Reperfusion Injury. Cell Metab. 23, 254–263.

We have added the recommended references as references 69 and 70 in the revised manuscript. We also added oxygenation of succinate as main superoxide generative species via reverse electron transport

“9. Sentence 248;
It is not clear which antioxidant enzymes are persulfidated?

We have removed the uncertain persulfidated antioxidant enzymes statement and rephrased the sentences.

“10. Sentence 307;
Add the following reference along the reference 83:

Zivanovic, J., Kouroussis, E., Kohl, J.B., Adhikari, B., Bursac, B., Schott-Roux, S., Petrovic, D., Miljkovic, J.L., Thomas-Lopez, D., Jung, Y., et al. (2019). Selective Persulfide Detection Reveals Evolutionarily Conserved Antiaging Effects of S-Sulfhydration. Cell Metab. 30, 1152-1170.e13.”

This has been added as reference 95 in the revised manuscript.

“11. Sentence 311; Check the reference 84!”

This has been corrected as reference 21 in the revised manuscript

Best,

Max

Reviewer 2 Report

In this manuscript “Hydrogen Sulfide Metabolite, Sodium Thiosulfate: Clinical 2 Applications and Underlying Molecular Mechanisms”, Dr. Sener review the mentalism, biological properties, current clinical usefulness and potential future clinical application of sodium thiosulfate. The manuscript is well-organized. Sulfur related therapeutic reagent development is a hot topic now and this review can serve as a good summary of the current and potential clinical application of sodium thiosulfate. There are several scientific descriptions need to be corrected and the language will need to be approved. I would suggest publication after the authors make the corresponding revisions.

  1. “Besides its endogenous production, H2S is also administered exogenously through a number of its donor compounds including STS, sodium hydrosulfide (NaHS), sodium sulfide (Na2S), diallyl disulfide (DADS), diallyl trisulfide (DATS), AP39 and GYY4137.” This would need a reference.
  2. “Contrary to this finding, other studies do not support the contribution of GSH in H2S production from thiosulfate. Olson et al. found that H2S generation from thiosulfate was due to the presence of 1,2-Dithiole-3-thiones, an exogenous reducing agent.” When it was found that thiosulfate would generate H2S in the presence of 1,2-Dithiole-3-thiones, it doesn’t mean of thiosulfate wouldn’t generate H2S in the presence of GSH. So, the statement that  “Olson et al. found that H2S generation from thiosulfate was due to the presence of 1,2-Dithiole-3-thiones, an exogenous reducing agent” cannot used as evidence that “other studies do not support the contribution of GSH in H2S production from thiosulfate”. Those two won’t contradict each other. So, this statement needs to be modified. From a chemical point of view, thiosulfate can generate H2S in the presence of any  free thiol species.
  3. “persulfidation” and “sulfhydration” are actually describing the same biological process: the thiol group (-SH) on the cysteine in protein was converted to a perthiol (-SSH) group. However, the authors described in a way as if they are different.  For example in those sentences:  “Many of the biochemical characteristics of H2S signaling that provide cytoprotective effects, such as persulfidation and sulfhydration of signaling proteins, can be accomplished with thiosulfate instead”  and “showed that H2S donor molecule GYY4137 inhibits Tau hyperphosphorylation by sulfhydrating  and persulfidating its kinase GSK3β,”. In the earlier days, people use protein sulfhydration more often. Recently, protein S-persulfidation are more widely used.  I would suggest to use protein S-persulfidation.
  4. For all the in vivo biological experiments, the following should be included: animal model, dosages and administration method. The authors include those key elements in Table1, however, the authors  didn’t to include those key elements in other parts of the manuscript.
  5. This statement is not right. “It is important to note that thiosulfate has two unpaired electrons: one at the single bonded sulfur moiety of the disulfide bond, and the other at the single bonded oxygen.” It is not “unpaired electrons”, it is called “lone electron pairs”
  6. “STS preconditioning significantly increased NADH dehydrogenase activity and decreased oxidative stress due to increased activities of antioxidant enzymes such as GSH, catalase, and superoxide dismutase”. GSH is not antioxidant enzymes
  7. “With H2S prodrugs being several decades away from clinical use”. This is not accurate. There are several H2S donors in clinic trials now such as SG1002, ATB-346 and Zofenopril. SG1002 are sold as a prescription medical food under the trade name ‘‘Sulfzix”.

The language needs significant improvement. I  only listed a few issues.

  1. “STS has several other uses including a common food preservative, a water dechlorinator, a photographic fixative, and a bleaching agent for paper pulp” should change to “STS has several other uses including as a common food preservative……”
  2. “Firstly, as illustrated in figure 1, a membrane-bound sulfide, quinone oxidoreductase (SQR), oxidizes H2S to persulfide, which is transferred to a persulfide group (SQR-SSH).” This sentence needs to be re-organized. I can understand what the authors are trying to say only because I am very familiar with this topic. This sentence itself is very confusing. One possible suggestion : “H2S can react with a membrane-bound disulfide on quinone oxidoreductase (SQR) to generate a membrane-bound persulfide group ”.
  3. “Although human mitochondria also utilize this sulfide oxidation pathway, recent evidence suggests that the persulfide intermediate  formed by human SQR uses glutathione (GSH)” This looks like an incomplete sentence and I am not quite sure what the authors are trying to say. It is the same case for “Most recently, Libiad et al. found that the kinetic behavior of these enzymes favors SQR using GSH as an acceptor to form glutathione persulfide (GSSH), which is then converted to thiosulfate by human 90 rhodanase (Figure 1).”
  4. Escherichia coli should be italic.
  5. This sentence is unclear. “In addition, Tokuda et al. observed increased plasma thiosulfate level and rhodanase activity when challenged by lipopolysaccharide (LPS) with or without H2S inhalation.”

Author Response

Please see the attachment: 

  1. Manuscript with track changes according to the reviewer's comments.

First of all, I would like to kindly thank the reviewers for their feedbacks on our manuscript and thereby, assisting in improving the quality of our work. Please find our responses to each comment below in dark blue text.

Reviewer #2

  1. “Besides its endogenous production, H2S is also administered exogenously through a number of its donor compounds including STS, sodium hydrosulfide (NaHS), sodium sulfide (Na2S), diallyl disulfide (DADS), diallyl trisulfide (DATS), AP39 and GYY4137.” This would need a reference.

Although NaHS, Na2S, DATS, DADS and AP39 are known H2S donors, we have decided to delete them from the list since they are not the focus of our work and were not mentioned anywhere else in the manuscript after being listed here. However, we maintained STS (the H2S donor of interest) and GYY4137, which was mentioned in the latter part of the manuscript, and provided references as requested [27-29].

“2. Contrary to this finding, other studies do not support the contribution of GSH in H2S production from thiosulfate. Olson et al. found that H2S generation from thiosulfate was due to the presence of 1,2-Dithiole-3-thiones, an exogenous reducing agent.” When it was found that thiosulfate would generate H2S in the presence of 1,2-Dithiole-3-thiones, it doesn’t mean of thiosulfate wouldn’t generate H2S in the presence of GSH. So, the statement that “Olson et al. found that H2S generation from thiosulfate was due to the presence of 1,2-Dithiole-3-thiones, an exogenous reducing agent” cannot used as evidence that “other studies do not support the contribution of GSH in H2S production from thiosulfate”. Those two won’t contradict each other. So, this statement needs to be modified. From a chemical point of view, thiosulfate can generate H2S in the presence of any free thiol species.”

As suggested by the reviewer, the statement has been modified

  1. Persulfidation” and “sulfhydration” are actually describing the same biological process: the thiol group (-SH) on the cysteine in protein was converted to a perthiol (-SSH) group. However, the authors described in a way as if they are different. For example in those sentences:  “Many of the biochemical characteristics of H2S signaling that provide cytoprotective effects, such as persulfidation and sulfhydration of signaling proteins, can be accomplished with thiosulfate instead”  and “showed that H2S donor molecule GYY4137 inhibits Tau hyperphosphorylation by sulfhydrating  and persulfidating its kinase GSK3β,”. In the earlier days, people used protein sulfhydration more often. Recently, protein S-persulfidation are more widely used.  I would suggest to use protein S-persulfidation.”

Removed sulfhydration from both sentences in lines x, y and only used persulfidation instead.

In addition, we replaced any use of the word “sulfhydration” with “persulfidation” in other parts of the manuscript.

“4. For all the in vivo biological experiments, the following should be included: animal model, dosages and administration method. The authors include those key elements in Table1, however, the authors didn’t to include those key elements in other parts of the manuscript.”

We have added those key elements of animal model and dosages to other parts of the revised manuscript.

“5. This statement is not right. “It is important to note that thiosulfate has two unpaired electrons: one at the single bonded sulfur moiety of the disulfide bond, and the other at the single bonded oxygen.” It is not “unpaired electrons”, it is called “lone electron pairs”

We agree with the reviewer. Hence, have corrected it to “lone electron pairs”

“6. STS preconditioning significantly increased NADH dehydrogenase activity and decreased oxidative stress due to increased activities of antioxidant enzymes such as GSH, catalase, and superoxide dismutase”. GSH is not antioxidant enzymes”

We have removed GSH since it is not antioxidant enzyme.

“7. With H2S prodrugs being several decades away from clinical use”. This is not accurate. There are several H2S donors in clinic trials now such as SG1002, ATB-346 and Zofenopril. SG1002 are sold as a prescription medical food under the trade name ‘‘Sulfzix”.

We agree with the reviewer. Therefore, we have corrected the statement by saying that “Considering that STS is already a clinically viable H2S donor drug approved by FDA and is also in clinical trials along with other H2S donor drugs such as GIG-1001, SG1002, ATB-436, and Zofenopril for cardiovascular diseases, intestinal disorders and other conditions…”

The language needs significant improvement. I only listed a few issues.

“1. STS has several other uses including a common food preservative, a water dechlorinator, a photographic fixative, and a bleaching agent for paper pulp” should change to “STS has several other uses including as a common food preservative……”

We have rephrased this sentence in the revised manuscript

“2. Firstly, as illustrated in figure 1, a membrane-bound sulfide, quinone oxidoreductase (SQR), oxidizes H2S to persulfide, which is transferred to a persulfide group (SQR-SSH).” This sentence needs to be re-organized. I can understand what the authors are trying to say only because I am very familiar with this topic. This sentence itself is very confusing. One possible suggestion: “H2S can react with a membrane-bound disulfide on quinone oxidoreductase (SQR) to generate a membrane-bound persulfide group”.

We have rephrased this sentence in the revised manuscript

“3. Although human mitochondria also utilize this sulfide oxidation pathway, recent evidence suggests that the persulfide intermediate formed by human SQR uses glutathione (GSH)” This looks like an incomplete sentence and I am not quite sure what the authors are trying to say. It is the same case for “Most recently, Libiad et al. found that the kinetic behavior of these enzymes favors SQR using GSH as an acceptor to form glutathione persulfide (GSSH), which is then converted to thiosulfate by human rhodanase (Figure 1).”

Sentence has been rephrased to make it clearer.

For the purpose of clarity, we have introduced the preposition “by” in the second sentence. Thus, the sentence now reads “Most recently, Libiad et al. found that the kinetic behavior of these enzymes favors SQR by using GSH as an acceptor to form glutathione persulfide (GSSH), which is then converted to thiosulfate by human rhodanase

“4. Escherichia coli should be italic.

We agree with the reviewer since it is a scientific name. Hence, E. coli has been italicized in the revised manuscript.

“5. This sentence is unclear. “In addition, Tokuda et al. observed increased plasma thiosulfate level and rhodanase activity when challenged by lipopolysaccharide (LPS) with or without H2S inhalation.”

We have rephrased the sentence to make it clearer.

Best,

Max
